# Weakly Supervised Understanding of Skilled Human Activity in Videos

## Abstract

Understanding skilled human activity is crucial in fields such as sports analytics, medical training, and professional development, where assessing proficiency can directly influence performance and outcomes. However, many existing approaches rely on human-annotated numerical scores or rankings, which are not only time-consuming but also introduce subjectivity. Conversely, categorizing proficiency as either high or low, though providing less detailed information, is easier to collect and can often be derived from group characteristics such as the distinction between novices and experts in surgical training. This new setting challenges models to uncover intrinsic patterns that reflect proficiency based solely on these weak labels. To achieve this, we introduce Sparse Skill Extractor, a multiscale contrastive learning framework. It enforces both local and global feature comparisons between groups while pruning irrelevant video segments to highlight key moments of skilled or unskilled performance. Our results demonstrate that Sparse Skill Extractor not only delivers strong performance in predicting demonstrator proficiency but also enhances interpretability by facilitating the detection of non-proficient timestamps for low proficiency demonstrations.

## 1 Introduction

The notion of skill is present across a wide variety of domains, ranging from cooking an omelet to executing a dive or performing a surgical procedure. Building models capable of perceiving skill enables the opportunity of automating feedback and providing real-time guidance, with significant potential applications in fields such as sports analytics (Pirsiavash et al., 2014; Bertasius et al., 2017; Parmar & Tran Morris, 2017; Parmar & Morris, 2019b) and surgical training (Ismail Fawaz et al., 2018; Zia et al., 2018; Liu et al., 2021).

Numerous works on action quality assessment (AQA) focus on predicting precise numerical scores in competitive sports, particularly the Olympics (Pirsiavash et al., 2014; Parmar & Morris, 2019b;a; Xu et al., 2022). While this setting is narrow, it is appealing because sports broadcast footage is readily accessible and includes detailed, systematically evaluated scores from judges (Pirsiavash et al., 2014). Nonetheless, skill is exhibited across a wide range of tasks, many of which do not naturally provide such precise numerical labels for model supervision. For these tasks, one approach is to develop "objective" scoring systems, as seen in surgical assessment (Martin et al., 1997; Vassiliou et al., 2005). However, achieving high inter-rater reliability (IRR) requires in-depth rater training and retraining after non-use (Robertson et al., 2018; Gawad et al., 2019). The alternative approach of ranking videos (Doughty et al., 2019; 2018; Malpani et al., 2014), while removing the need to create a numerical scoring system, demands extensive annotation collection. For instance, the Bristol Everyday Skill Tasks dataset collected 16,782 paired annotations to rank five tasks each including 100 videos (Doughty et al., 2019).

In contrast, our work explores the efficacy of understanding skilled human activity using only binary labels of high or low demonstrator proficiency. In this setting, annotations are (1) easier to collect, (2) less prone to subjectivity due to their coarser nature, and (3) can even be acquired without annotating labels in tasks where inherent expert-novice distinctions exist, such as surgical training or sports coaching.

Still, predicting proficiency based on binary labels remains challenging. It requires a detailed understanding of how specific steps are executed and how subtle aspects of task performance contribute

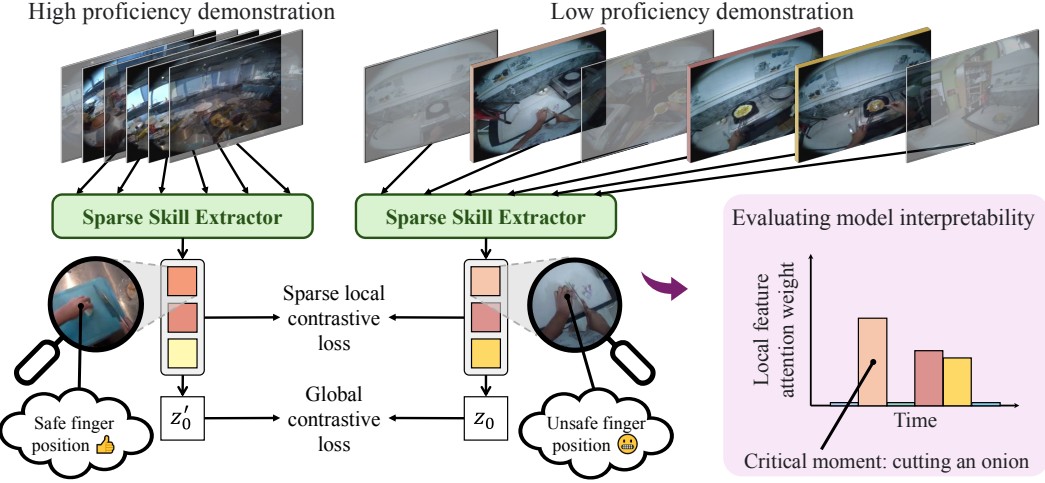

Figure 1: **Overview of our work**. Our proposed method utilizes binary proficiency labels to generate sparse representations that retain only the video segments relevant to proficiency, such as those demonstrating knife skills during chopping. The model is trained using a contrastive loss applied to both the sparse local segment features and the generated global feature. We evaluate model interpretability by examining whether the flow of information used to produce the global feature is greater for segments that contain critical moments indicating proficiency. The gray video frames represent segments irrelevant to proficiency which get pruned away by the Sparse Skill Extractor framework.

to overall proficiency. The difficulty is magnified in long-form tasks, where skill may only be expressed during key moments. A straightforward approach to learning skill proficiency from binary labels involves contrastive learning using either a feature from every video segment or a global feature from the entire video. However, this approach makes the naive assumption that skill expression is uniform throughout the video, implying that all parts of a video are indicative of skill. In reality, this is rarely the case. For instance, in a demonstration of cooking an omelet, proficiency may only become apparent in key moments, such as how the vegetables are chopped or the omelet is flipped. Thus, for models to achieve reliable performance, they must be interpretable by identifying the key moments that affect the proficiency score.

To achieve this interpretability, we introduce a multiscale contrastive learning framework that focuses specifically on the moments most relevant to proficiency. Our proposed approach, Sparse Skill Extractor, first extracts features from individual video segments, then selectively prunes the segments and applies a contrastive loss only to: (1) sparse local features from the remaining segments, and (2) a global feature generated through sparse self-attention of the remaining local features (see Figure 1). This approach allows us to precisely identify critical moments indicative of proficiency, even in long-form videos containing many steps.

We evaluate our method on the challenging dataset of Ego-Exo4D (Grauman et al., 2023), which contains long-form videos of procedural cooking tasks. Compared to baselines and ablations, our Sparse Skill Extractor framework demonstrates strong performance in predicting demonstrator proficiency and enhances interpretability as measured through analysis of the model's attention weights. Additionally, we extend our evaluation to popular AQA datasets, FineDiving (Xu et al., 2022) and JIGSAWS (Gao et al., 2014), to assess how well our method can infer precise proficiency scores using only binary labels as supervision in contrast to fully supervised baseline approaches trained with numerical labels. Despite requiring significantly less supervision, our method achieves performance approaching that of fully supervised methods, highlighting the effectiveness of our approach in learning robust features for skilled human activity understanding. Finally, we explore the use of skill experience (expert vs. novice) as a proxy for proficiency in the JIGSAWS dataset. Our findings show that using these inherent characteristics as supervision yields comparable results to using annotated labels for predicting numerical proficiency, further demonstrating the utility of our setup.

## 2 METHODS

In this section, we introduce Sparse Skill Extractor as a method that utilizes binary proficiency as weak labels to learn a fine-grained understanding of skilled human activity through contrastive learning of sparse local and global features. In Section 2.1, we overview the problem formulation. In Section 2.2, we present our proposed framework.

### 2.1 PROBLEM FORMULATION

Using only binary proficiency labels as supervision, the goal of our work is to learn robust representations that can both accurately predict binary proficiency and extrapolate to fine-grained numerical scores, while ensuring model interpretability by attending to key moments in the video that indicate proficiency. Formally, given a video $\mathcal{V}$ with binary proficiency label $y$, we aim to learn a representation $z_0$ that satisfies the following three criteria: (1) a linear classifier attached to $z_0$ accurately predicts $y$; (2) the predicted probability of $y$ from the linear classifier is discriminative for numerical proficiency evaluation, outputting a greater probability of high proficiency to a demonstration with a higher numerical proficiency score, even when comparing two demonstrations with the same binary proficiency label; and (3) assuming $z_0$ is generated from a Transformer architecture, the quantified flow of information to generate $z_0$ (denoted as $\tilde{\mathbf{A}}_{0,1:N} \in \mathbb{R}^N$, where there are $N$ segments in $\mathcal{V}$) is higher for segments containing critical moments of proficiency compared to non-critical segments. The metrics used to evaluate these criteria are detailed in Section 3.1.

### 2.2 SPARSE SKILL EXTRACTOR

In order to obtain such representations from course binary proficiency labels, we follow a contrastive learning setup. To this effect, we formulate the training paradigm as comparing two different video demonstrations of the same task. Namely, given a query video $\mathcal{V}$ with binary proficiency label $y$ and a randomly sampled comparison video $\mathcal{V}'$ with binary proficiency label $y'$, the task is to generate $z_0$ and $z_0'$ which are similar if $y = y'$ and dissimilar otherwise. In order to generate $z_0$ and $z_0'$, we first split $\mathcal{V}$ and $\mathcal{V}'$ into $N$ segments and encode each segment to obtain local segment features. To avoid making the naive assumption that skill expression is uniform throughout the video, when generating $z_0$ and $z_0'$ from the local segment features, we employ a token sparsification module $\phi_{sparse}$ that filters out local segments not informative of skill. During training, we apply a contrastive objective on both the global features $z_0$ and $z_0'$ as well as the remaining, informative local segment features. We present an overview of this framework in Figure 1 and provide a more in-depth visualization in the supplement. Below, we detail each part of Sparse Skill Extractor.

**Video segment feature extraction.** We first split the query video $\mathcal{V}$ into $N$ partitions and randomly sample a segment of $K$ frames from each partition with a temporal stride of $f$ between sampled frames (denoted $\mathbf{v} = [v_1, ..., v_N]$ where $v_i$ includes $K$ frames $\forall i \in \{1, ..., N\}$). For each segment's starting frame in $\mathbf{v}$, we find the corresponding frame in the comparison video using a linear mapping and sample $K$ frames starting at this frame with the same temporal stride as the query video (denoted $\mathbf{v}' = [v_1', ..., v_N']$ where $v_i'$ includes $K$ frames $\forall i \in \{1, ..., N\}$). We then feed both the query video segments and comparison video segments through a pre-trained video encoder to obtain segment features $\mathbf{x} = [x_1, ..., x_N]$ and $\mathbf{x}' = [x_1', ..., x_N']$. Our framework does not depend on the specific type of video encoder, and in our experiments, we use various encoders.

**Generating sparse video representations.** From the generated video segment features $\mathbf{x}$ and $\mathbf{x}'$, we aim to construct video-wide representations for discerning proficiency. Since proficiency is likely demonstrated only at specific moments, we want the model to focus on the segments containing critical moments while ignoring the irrelevant segments. To achieve this, we employ a Transformer encoder with a token sparsification module $\phi_{sparse}$ between its two layers that drops uninformative tokens. Analogous to how Vision Transformer explainability approaches such as Rao et al. (2021) utilize a module to drop uninformative image patches, we use $\phi_{sparse}$ to filter out uninformative video segments. Specifically, $\phi_{sparse}$ is a light-weight module that takes as input the intermediate video segment tokens and updates a decision mask $\hat{\mathbf{D}}$ (all elements initialized to 1) that indicates whether to drop or keep each token. Given the first Transformer layer output of the query video, denoted $\mathbf{w}^1 = [w_1^1, ..., w_N^1]$ (excluding the [cls] token), we compute embeddings $\mathbf{u} = [u_1, ..., u_N]$

as a concatenation of local and global information:

$$u_i = [u_i^{\text{local}}, u_i^{\text{global}}], \quad 1 \leq i \leq N. \tag{1}$$

The local and global embeddings ($\mathbf{u}^{\text{local}} = [u_1^{\text{local}}, ..., u_N^{\text{local}}]$ and $\mathbf{u}^{\text{global}} = [u_1^{\text{global}}, ..., u_N^{\text{global}}]$) are generated as $\mathbf{u}^{\text{local}} = \text{MLP}(\mathbf{w}^1)$ and $\mathbf{u}^{\text{global}} = \text{Avg}(\text{MLP}(\mathbf{w}^1))$, where the same MLP is used for local and global embeddings and Avg is average pooling. In this way, each token's embedding contains information from its specific segment and context from the whole video. From here, we generate the decision mask:

$$\pi = \text{Softmax}(\text{MLP}(\mathbf{u})), \tag{2}$$

$$\hat{\mathbf{D}} = \text{Gumbel-Softmax}(\pi)_{*,1}, \tag{3}$$

where we use a separate MLP from the embedding generation, and we take index 1 of the Gumbel-Softmax as it represents the mask of the kept tokens.

To perform parallel training with the decision mask that can have a various number of kept tokens within a batch, we utilize the attention masking strategy. Namely, we calculate the self-attention matrix $\mathbf{A}$ by:

$$\mathbf{P} = \mathbf{Q}\mathbf{K}^T/\sqrt{d}, \tag{4}$$

$$\mathbf{G}_{ij} = \begin{cases} 1, & i = j, \\ \hat{\mathbf{D}}_j, & i \neq j. \end{cases} \quad 1 \leq i, j \leq N, \tag{5}$$

$$\mathbf{A}_{ij} = \frac{\exp(\mathbf{P}_{ij})\mathbf{G}_{ij}}{\sum_{k=1}^{N} \exp(\mathbf{P}_{ik})\mathbf{G}_{ik}}, \quad 1 \leq i, j \leq N, \tag{6}$$

where $d$ is the dimension of the segment features. Note that $\mathbf{A}$ is equivalent to the standard attention matrix by considering only the kept tokens, and a self-loop is added in $\mathbf{G}$ to improve numerical stability.

Once the output of the second Transformer layer is generated using this attention masking strategy for the query video $\mathbf{w}^2 = [w_0^2, w_1^2, ..., w_N^2]$ and comparison video $\mathbf{w}^{2\prime} = [w_0^{2\prime}, w_1^{2\prime}, ..., w_N^{2\prime}]$ (including the [cls] token), we apply a final MLP to the global features $w_0^2$ and $w_0^{2\prime}$ to obtain $z_0$ and $z_0'$ and the identify function to the local features $[w_1^2, ..., w_N^2]$ and $[w_1^{2\prime}, ..., w_N^{2\prime}]$ to obtain $[z_1, ..., z_N]$ and $[z_1', ..., z_N']$, respectively. For binary demonstrator proficiency prediction, we attach a linear classifier to $z_0$ to obtain $\hat{y}$. Note that we add a stop_gradient to the input of the linear classifier to prevent the binary classification prediction from influencing the learned representations. We do not predict the demonstration proficiency of the comparison video.

**Training and inference.** To enforce a contrastive objective on the global features $z_0$ and $z_0'$, we compute the global contrastive loss $\mathcal{L}_{global}$ as:

$$\mathcal{L}_{global} = \mathbb{1}\{y = y'\} \cdot \log \sigma(z_0 z_0') + \mathbb{1}\{y \neq y'\} \cdot \log(1 - \sigma(z_0 z_0')). \tag{7}$$

Additionally, for the informative local segment features remaining after sparsification $\{z_i \mid \hat{\mathbf{D}}_i = 1, 1 \leq i \leq N\}$, we calculate the local contrastive loss $\mathcal{L}_{local}$ as:

$$\mathcal{L}_{local} = \frac{1}{\sum_{i=1}^{N} \hat{\mathbf{D}}_i} \sum_{i=1}^{N} \mathbb{1}\{y = y'\} \cdot \hat{\mathbf{D}}_i \cdot \log \sigma(z_i z_i') + \mathbb{1}\{y \neq y'\} \cdot \hat{\mathbf{D}}_i \cdot \log(1 - \sigma(z_i z_i')). \tag{8}$$

To constrain the ratio of kept tokens in the sparsification module, we calculate the ratio loss $\mathcal{L}_{ratio}$ as the following MSE objective:

$$\mathcal{L}_{ratio} = \left( \mu - \frac{1}{N} \sum_{i=1}^{N} \hat{\mathbf{D}}_i \right)^2, \tag{9}$$

where $\mu$ is the predefined target ratio.

Lastly, we calculate the classification loss $\mathcal{L}_{class}$ as:

$$\mathcal{L}_{class} = -(y \cdot \log \hat{y} + (1 - y) \cdot \log(1 - \hat{y})). \tag{10}$$

Our overall loss is a combination of the individual losses:

$$\mathcal{L} = \mathcal{L}_{global} + \mathcal{L}_{local} + \mathcal{L}_{ratio} + \mathcal{L}_{class}. \tag{11}$$

During inference, we sample 10 comparison videos for each query video.

## 3 EXPERIMENTS

In this section, we assess the efficacy of our Sparse Skill Extractor framework in learning robust, interpretable representations that are predictive of skill proficiency. We first introduce the metrics, datasets, baselines, and implementation details. We then present the results and analyses.

### 3.1 EVALUATION METRICS

Below, we overview the metrics used to evaluate the effectiveness of satisfying the three criteria outlined in Section 2.1.

**Binary $F_1$.** To evaluate binary proficiency prediction performance, we use $F_1$ score. Note that this metric is only used for models that generate binary predictions (excluding fully supervised baselines trained using numerical proficiency scores).

**Spearman's rank correlation.** To measure the numerical proficiency prediction performance of fully supervised baselines and assess how effectively the weakly supervised methods extrapolate numerical scores from binary predictions, we use Spearman's rank correlation ($\rho$).

$$\rho = \frac{\sum_{j=1}^{M}(s_j - \overline{s})(\hat{s}_j - \overline{\hat{s}})}{\sqrt{\sum_{j=1}^{M}(s_j - \overline{s})^2 \sum_{j=1}^{M}(\hat{s}_j - \overline{\hat{s}})^2}}, \tag{12}$$

where $\mathbf{s}$ and $\hat{\mathbf{s}}$ denote the ranking of two series, respectively. For weakly supervised approaches, $\hat{\mathbf{s}}$ is ranked based on probabilities of high proficiency. For fully supervised methods, $\hat{\mathbf{s}}$ is ranked based on the predicted numerical scores. In all cases, $\mathbf{s}$ comprises the ranking of ground-truth numerical proficiency labels.

**Error Recall.** To quantify model interpretability, we evaluate whether the flow of information used for predictions in low proficiency demonstrations is greater in segments containing annotated errors compared to non-error segments. We measure this error-grounded behavior using recall. Formally, given a low proficiency demonstration with $\ell$ error steps (defined as $\mathbf{e} = \{e_i, 1 <= i <= \ell\}$), we measure recall as $|\mathbf{e} \cap \hat{\mathbf{e}}|/\ell$ where $\hat{\mathbf{e}}$ is comprised of the $\ell$ steps with the highest average model attention. To generate $\hat{\mathbf{e}}$, we use $\mathbf{A}^1$ and $\mathbf{A}^2$, the self-attention matrices from the first and second layer of the Transformer, respectively. Applying attention rollout (Abnar & Zuidema, 2020), we calculate the overall attention as $\tilde{\mathbf{A}} = \mathbf{A}^1\mathbf{A}^2$. We then select the weights from the global feature to get $\tilde{\mathbf{A}}_{0,1:N}$, representing the importance of each segment on demonstrator proficiency prediction. For each segment consisting of $K$ frames, we utilize annotated per-frame step labels to save the attention weights corresponding to each step. Finally, we define $\hat{\mathbf{e}}$ as the $\ell$ steps with the highest average attention weight.

### 3.2 DATASETS

**Ego-Exo4D (Cooking).** Our analysis centers on Ego-Exo4D (Grauman et al., 2023), a dataset containing skilled human activities in the challenging setting of long-form videos. We exclude non-procedural domains and examples without demonstrator proficiency annotations, narrowing our scope to cooking. We focus on procedural tasks to ensure that all examples within a task involve the same sequence of steps, requiring the model to discern fine-grained details of *how* steps are executed, rather than allowing skill to be inferred from outcomes such as reaching the top of a rock climbing wall or successfully making basketball shots. To ensure sufficient training data for each cooking task, we exclude tasks with less than 10 examples, resulting in a final selection of eight tasks (Cooking an Omelet, Cooking Tomato & Eggs, Cooking Scrambled Eggs, Making Cucumber & Tomato Salad, Making Sesame-Ginger Asian Salad, Cooking Noodles, Making Milk Tea, and Making Coffee Latte). In our work, we utilize the egocentric viewpoint as input to the model. We derive binary proficiency scores from the expert commentaries rating each example on a scale from 1 (least skilled) to 10 (most skilled), using a threshold of 4 to separate low and high proficiency. As the Ego-Exo4D test set is withheld, we use the official validation set as the test set and set aside 20% of examples from each task in the train set to use as the validation set for model selection.

Additionally, we annotate which steps contain errors in low proficiency demonstration to evaluate model interpretability. Although Ego-Exo4D includes timestamped comments from experts noting

good executions and mistakes, we do not use these annotations as they are collected after proficiency scoring and do not necessarily relate to the explanations given for demonstrator proficiency scores. Instead, we use the score explanations to manually select each step relevant to the explanations. We exclude examples where expert explanations do not directly refer to procedural steps. Examples of the generated error step annotations are present in Figure 2.

**FineDiving.** The FineDiving dataset (Xu et al., 2022) is a prevalent procedure-based AQA dataset containing videos of Olympic dives and numerical judge scores. We use this dataset to evaluate how well our weakly supervised method extrapolates fine-grained numerical scores compared to state-of-the-art fully supervised methods trained on numerical proficiency scores. To train our weakly supervised approach, we derive binary proficiency scores from the numerical dive scores thresholding based on the mean score for each dive. For our training setup, we exclude dives that contain less than 10 examples.

**JIGSAWS.** The JIGSAWS dataset (Gao et al., 2014) contains videos of surgical activities performed using the *da Vinci* Surgical System (Salisbury & Guthart, 2000) along with both global rating proficiency scores and expertise labels. With this dataset, we explore the potential of our approach for surgical applications and the ability to utilize experience as an inherent binary characteristic for supervision. Since the suturing and needle-passing tasks do not exhibit statistically significant correlation between proficiency and experience (Lefor et al., 2020), we only evaluate on the knot-tying task. We adopt the four-fold cross-validation splits from baseline approaches (Tang et al., 2020; Yu et al., 2021; Bai et al., 2022). In order to binarize the expertise labels, we combine the intermediate and expert classes.

Information about the dataset statistics is provided in Table 1. See the supplement for more details about dataset statistics.

Table 1: Statistics of the Ego-Exo4D (Cooking) (Grauman et al., 2023), FineDiving (Xu et al., 2022), and JIGSAWS (Gao et al., 2014) datasets

| Dataset | # Samples | # Tasks | Average Duration |
|---|---|---|---|
| Ego-Exo4D (Cooking) | 283 | 8 | 9.91m |
| FineDiving | 2918 | 32 | 8.68s |
| JIGSAWS | 103 | 3 | 1.54m |

### 3.3 BASELINES

We compare our method against numerous baselines, comprising weakly supervised baselines, ablations, and fully supervised baselines.

**Weakly supervised baselines.** We provide a series of weakly supervised baselines to evaluate the performance of approaches trained with the same level of supervision as our method. Similar to the methods presented for demonstrator proficiency estimation in Ego-Exo4D (Grauman et al., 2023), we employ various video encoders to predict binary proficiency from individual video segments. For our video encoders, we choose TimeSformer (Bertasius et al., 2021), which has demonstrated effectiveness in human activity understanding benchmarks, and the self-supervised V-JEPA architecture (Bardes et al., 2024), emerging as a strong model for video representation learning. We combine individual segment predictions by summing the logits prior to applying the cross-entropy loss and only utilize the egocentric view to more closely align with our method's training setup.

**Ablations.** In addition to the weakly supervised baselines, we also ablate our framework to determine the contributions of various components, including the token sparsification module, local contrastive loss, and global contrastive loss.

- **With non-sparse local contrastive loss** removes the token sparsification module and ratio loss (Eq. 9), instead enforcing the local contrastive loss (Eq. 8) on every token.

- **Without sparse local contrastive loss** both removes the token sparsification module and ratio loss (Eq. 9) and also removes the the local contrastive loss (Eq. 8).

- **Without local or global contrastive loss** does not have the token sparsification module and only uses the classification loss as the objective. Note that the stop gradient is removed to train more than the final MLP.

**Fully supervised baselines.** To illustrate the effectiveness of our weakly supervised method in extrapolating numerical proficiency scores using only binary labels, we compare it against various fully supervised AQA methods that use numerical proficiency labels for supervision. These methods include Uncertainty-aware Score Distribution Learning (USDL) and Multi-path Uncertainty-aware Score Distributions Learning (MUSDL) (Tang et al., 2020), Contrastive Regression (CoRe) (Yu et al., 2021), Multi-stage Contrastive Regression (MCoRe) (An et al., 2024), Temporal Segmentation Attention (TSA) (Xu et al., 2022), and Temporal Parsing Transformer (TPT) (Bai et al., 2022). All of these methods utilize precise, numeric proficiency labels during training, and mCoRe and TSA also use step transition annotations during training.

## 3.4 IMPLEMENTATION DETAILS

When using the token sparsification module, we always set the target ratio, $\mu$, to 0.5. For our Ego-Exo4D (Cooking) experiments, we used the pre-trained V-JEPA model (Bardes et al., 2024) with the ViT-L/16 architecture as the video encoder. Following the attentive probing protocol of V-JEPA, we kept the backbone frozen and employed a learnable non-linear pooling strategy consisting of a cross-attention layer with a learnable query token which is then added back to the query token and fed into a two-layer MLP followed by a LayerNorm (without the last linear layer used for classification). The V-JEPA weakly supervised baseline utilized this same setup. For the TimeSformer weakly supervised baselines (Bertasius et al., 2021), we experimented with models pre-trained on the K400 and HowTo100M datasets and froze the entire backbone. For FineDiving and JIGSAWS experiments, to maintain consistency with the experimental setup of fully supervised baseline methods (Tang et al., 2020; Yu et al., 2021; Xu et al., 2022; Bai et al., 2022), we utilized the I3D model pre-trained on Kinetics (Carreira & Zisserman, 2017) as the video encoder and fine-tuned the entire backbone. Additional implementation details are available in the supplement.

## 3.5 SPARSE SKILL EXTRACTOR OUTPERFORMS WEAKLY SUPERVISED BASELINES

Table 2: Results on the long-form Ego-Exo4D (Cooking) dataset (Grauman et al., 2023) compared to weakly supervised baselines and ablations. Performance is evaluated using $F_1$ score for binary demonstrator proficiency prediction, Spearman's correlation for exact, numerical proficiency prediction, and recall for error detection. For each metric, the best performance is **bolded** and the second best is underlined. Note that error recall performance is not measurable for weakly supervised baselines as they follow a late-fusion approach.

| Method | Binary $F_1$ | Sp. Corr. | Error Recall |
|---|---|---|---|
| Random | 0.317 | 0.075 | 0.094 |
| TimeSformer (K400) (Bertasius et al., 2021) | 0.523 | 0.016 | – |
| TimeSformer (HowTo100M) (Bertasius et al., 2021) | 0.468 | -0.057 | – |
| V-JEPA (Bardes et al., 2024) | 0.591 | 0.196 | – |
| Sparse Skill Extractor (Ours) | 0.618 | 0.485 | **0.365** |
| w/ non-sparse local contrastive loss | **0.621** | **0.491** | 0.292 |
| w/o sparse local contrastive loss | 0.559 | 0.261 | 0.323 |
| w/o local or global contrastive loss | 0.591 | 0.170 | 0.302 |

We first compare our method with weakly supervised baselines and ablations on the challenging, long-form Ego-Exo4D (Cooking) dataset. We provide full results in Table 2. For binary proficiency prediction (measured using $F_1$) and extrapolated numerical prediction (measured using Spearman's correlation), we find that our Sparse Skill Extractor method yields strong performance, almost matching the setup that enforces contrastive loss on all local features ($\Delta - 0.03$ on Binary $F_1$ and $\Delta - 0.06$ on Sp. Corr.) despite our approach not using all video segment features for proficiency prediction (as segments are pruned by $\phi_{sparse}$). This finding indicates that the pruned video

segments contribute little to proficiency prediction, validating the effectiveness of the sparsification module in removing segments not informative of skill. Meanwhile, we find that our method greatly outperforms all other compared approaches. Particularly for numerical proficiency prediction, we see substantial drops in performance when removing the sparse local contrastive loss ($\Delta - 0.224$ Sp. Corr.) and local + global contrastive losses ($\Delta - 0.315$ Sp. Corr.). This finding demonstrates the importance of enforcing a constrastive loss on the local video segment features to learn a nuanced understanding of skill proficiency. Analyzing the proficiency prediction performance of TimeSformer and V-JEPA, we find that while both models achieve reasonable binary prediction performance (though less effectively than our approach), their performance largely decreases when extrapolating to fine-grained numerical scores. This result highlights the difficulty of learning representations that effectively discriminate numerical proficiency from binary supervision, particularly when using a late fusion setup that assumes all segments contribute equally to proficiency.

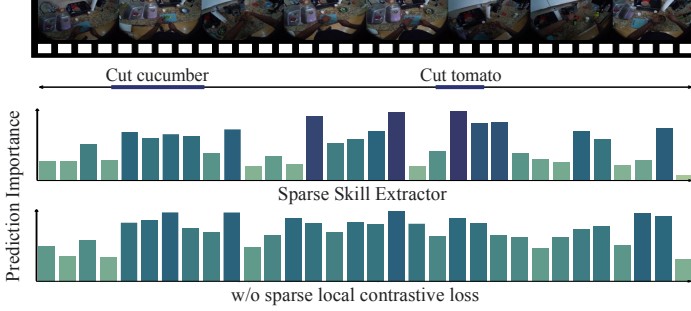

Figure 2: **Visualizing model interpretability for low proficiency videos.** Each example includes the expert commentary used to extract step error annotations on the left, extracted step errors below the video, and attention weights leading to proficiency prediction for both our approach and the approach without the local contrastive loss. We find that our sparse local contrastive loss leads to less uniform attention weights with a higher focus on error regions compared to the approach without a sparse local contrastive loss.

When evaluating model interpretability with error recall, we find that our method with the sparse local loss performs much better than all other setups ($\Delta + 0.042$ second-best setup). Of note, the setup with a non-sparse local contrastive loss achieves the worst performance, likely because this approach makes the assumption that skill expression is uniform through the video. See Figure 2 for qualitative examples comparing error detection abilities of our approach and the second-best setup excluding the local contrastive loss and token sparsification module.

### 3.6 SPARSE SKILL EXTRACTOR APPROACHING FULLY SUPERVISED PERFORMANCE

We additionally compare our weakly supervised method only using binary demonstrator proficiency labels to fully supervised approaches that use exact, numerical scores on the FineDiving (Xu et al., 2022) and JIGSAWS (Gao et al., 2014) datasets. On FineDiving, we find that our approach achieves a Spearman's correlation of 0.7478, whereas the fully supervised methods achieve performances

between 0.8302 and 0.9232. Note that the highest-performing methods additionally utilize step transition annotations during training. Although our approach does not reach the same performance as fully supervised methods, it still achieves promising results given the weakly supervised setting. We see a similar pattern for the JIGSAWS dataset, where our model achieves a binary demonstrator proficiency $F_1$ score of 0.835 and Spearman's correlation of 0.65. Table 3 shows full results comparing our approach to fully supervised methods.

Table 3: Results on the FineDiving (Xu et al., 2022) and JIGSAWS (Gao et al., 2014) datasets. For weakly supervised results, FineDiving only includes tasks with at least 10 dives and Sp. Corr. is calculated within each event type and the average across events is taken. $^\ddagger$ indicates using step transition annotations during training. For each metric, the best fully supervised performance is boxed and the best weakly supervised performance is **bolded**. Note that fully supervised baselines do not generate binary proficiency predictions.

| Method | FineDiving | | JIGSAWS (Knot-Tying) | |
|---|---|---|---|---|
| | Binary $F_1$ | Sp. Corr. | Binary $F_1$ | Sp. Corr. |
| *Fully Supervised* | | | | |
| USDL (Tang et al., 2020) | – | 0.8302 | – | 0.61 |
| MUSDL (Tang et al., 2020) | – | 0.8427 | – | 0.71 |
| CoRe (Yu et al., 2021) | – | 0.9061 | – | 0.86 |
| mCoRe$^\ddagger$ (An et al., 2024) | – | 0.9232 | – | – |
| TSA$^\ddagger$ (Xu et al., 2022) | – | 0.9203 | – | – |
| TPT (Bai et al., 2022) | – | – | – | 0.91 |
| *Weakly Supervised* | | | | |
| Sparse Skill Extractor (Ours) | **0.779** | **0.7478** | **0.835** | 0.65 |
| w/o local or global contrastive loss | 0.762 | 0.7022 | 0.704 | **0.77** |
| w/ experience supervision | – | – | 0.555 | 0.65 |
| Random | 0.023 | 0.2828 | 0.431 | 0.23 |

### 3.7 CASE STUDY: EXPERIENCE AS AN EFFECTIVE PROXY FOR PROFICIENCY

Given that one advantage of training with binary labels is that the natural distinction between experts and novices can be leveraged for supervision without requiring raters for label collection, we explore the effectiveness of using experience as a proxy for proficiency. On the JIGSAWS knot-tying task, we observe that while training with experience labels leads to a decrease in binary proficiency prediction performance compared to using proficiency labels ($\Delta - 0.280$ on Binary $F_1$), it matches the performance for extrapolated numerical proficiency prediction. This indicates that, even when utilizing self-reported experience labels as supervision, our approach can still learn useful representations for distinguishing numerical proficiency.

## 4 RELATED WORK

**Understanding skilled human activity.** Skill assessment is a growing area of interest across many domains such as surgical tasks (Ismail Fawaz et al., 2018; Zia et al., 2018; Liu et al., 2021) and sports (Pirsiavash et al., 2014; Bertasius et al., 2017; Parmar & Tran Morris, 2017; Parmar & Morris, 2019b). Traditionally, AQA is formulated as a regression task based on numerical score labels provided by task experts (Parmar & Tran Morris, 2017; Parmar & Morris, 2019b;a). The first work to propose a generic learning-based framework for AQA extracted spatio-temporal pose features for Olympic score prediction (Pirsiavash et al., 2014). Popular datasets for regression-based AQA include AQA-7 (1106 action samples from Summer and Winter Olympics) (Parmar & Morris, 2019a), MTL-AQA (1412 diving samples) (Parmar & Morris, 2019b), FineDiving (3000 diving samples) (Xu et al., 2022), and JIGSAWS (103 surgical activity samples) (Gao et al., 2014).

Related to our approach, a series of works explore how to utilize information across various stages of video demonstrations to learn fine-grained proficiency scores. For example, Xu et al. (2022)

generate procedure-aware embeddings by first parsing actions into consecutive steps with semantic and temporal correspondences. Similarly, Huang & Li (2024) segment features into a semantic sequence. However, these approaches rely on step transition labels to learn temporal information. Moving beyond the supervised setting to learn temporal information, Roditakis et al. (2021) concatenate appearance features with self-supervised features based on video alignment to improve AQA performance. Likewise, Bai et al. (2022) introduce temporal alignment in a self-supervised fashion with a temporal parsing transformer to decompose holistic features into temporal part-level representations. In our work, we enable precise discrimination of proficiency in critical execution steps by incorporating a sparse local contrastive loss. The Ego-Exo4D dataset (Grauman et al., 2023) offers detailed explanations for proficiency scores, providing a valuable resource for assessing model interpretability by evaluating how these critical execution steps are utilized for proficiency prediction.

In addition to the regression formulation, a series of works look at formulating the problem as a pairwise ranking of skill between two videos (Doughty et al., 2019; 2018; Malpani et al., 2014). A recent work goes beyond the traditional pairwise ranking and incorporates an expert demonstration video as a reference point (Huang et al., 2024). This work additionally includes both egocentric and exocentric views of the demonstration. There are few but limited works that use the expert-novice distinction as supervision for training networks. For example, a series of studies explore experience prediction on the JIGSAWS dataset (Soleymani et al., 2021; Nguyen et al., 2019; Funke et al., 2019). However, these works do not explore extrapolating to numerical scores or assessing model interpretability.

**Transformer sparse feature learning.** In recent years, there have been a series of works studying the pruning of Vision Transformers to improve model efficiency and interpretability. For example, Pan et al. (2021) propose multi-head interpreters that drop uninformative patches and are optimized by a reward that balances efficiency and accuracy. Yu & Xiang (2023) improve explainability by creating a mask that measures unit (e.g., attention heads or matrices in linear layers) contribution to the predicting of each target class and only preserving the most informative units. Liang et al. (2022) improve efficiency by fusing inattentive tokens in order to speed up subsequent attention and feed-forward computations. Rao et al. (2021) prune tokens by using a lightweight prediction module to estimate the importance of each token. In our work, rather than simply removing redundant visual patches to enhance efficiency, we focus on identifying and pruning entire video features that are irrelevant to proficiency. This approach enables us to learn more interpretable and robust features for skilled human activity understanding.

## 5 CONCLUSION

In this work, we investigate the efficacy of utilizing binary proficiency labels as weak supervision for learning robust skill-based representations. Motivated by the challenges of this setup, we propose the Sparse Skill Extractor, which focuses specifically on the moments most relevant to proficiency. Our results demonstrate that our proposed framework not only excels in predicting binary proficiency but also effectively extrapolates to numerical proficiency prediction while enhancing model interpretability.

Our work reveals that binary proficiency supervision holds significant potential for efficiently developing models with a nuanced understanding of skill. Future work may explore several exciting directions, such as scaling up data for more generalizable representations, leveraging other binary labels reflective of skill for supervision such as observed patient outcomes from surgical procedures, and advancing toward the automation of feedback by leveraging the critical moments of proficiency identified by our framework to explain the differences between low and high proficiency execution using natural language.

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
