## A    SUPPLEMENTARY MATERIAL FOR WEAKLY SUPERVISED UNDERSTANDING OF SKILLED HUMAN ACTIVITY IN VIDEOS

### A.1    FRAMEWORK VISUALIZATION

We present an in-depth visualization of our proposed approach in Figure 3.

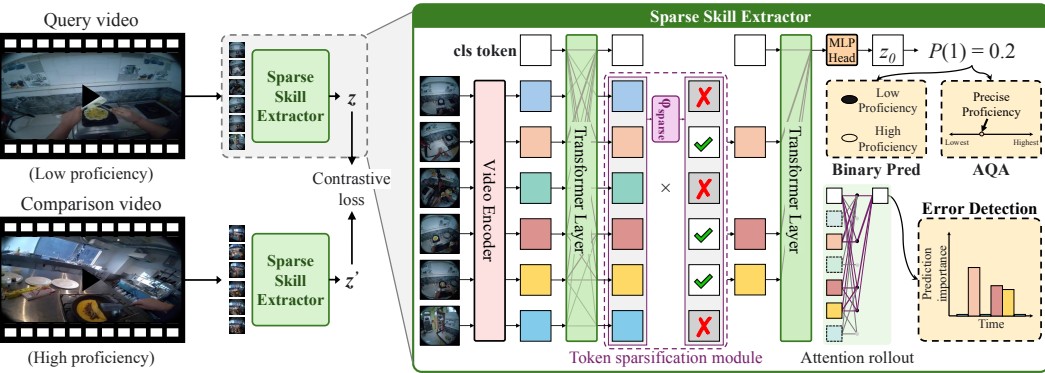

Figure 3: **Framework of the proposed approach.** Given query and comparison videos, they are partitioned into segments and features are extracted using a video encoder. These video segment representations are then passed through a shallow Transformer encoder, where a token sparsification module is inserted between layers to prune segments of the query video that are less informative of skill. A contrastive loss is enforced between the query and comparison video over the final Transformer outputs of the remaining video segment (local) tokens and [cls] (global) token. A classification head is attached to the global token output to predict demonstrator proficiency. Through attention rollout of the Transformer (Abnar & Zuidema, 2020), the most salient segments for skill prediction of low proficiency query videos are retrieved and their efficacy is evaluated via error detection.

### A.2    ADDITIONAL IMPLEMENTATION DETAILS

For the Ego-Exo4D (Cooking) and JIGSAWS datasets consisting of longer-form videos, we set the number of video partitions ($N$) as 32, frames per segment ($K$) as 16, and temporal stride ($f$) as 4. For the shorter-form FineDiving dataset, we set $N = 4$, $K = 4$, and $f = 1$. The only exception to this was the TimeSformer setup on Ego-Exo4D, for which we set $N = 8$, $K = 8$, and $f = 32$. This configuration was chosen to maintain the same frame sampling strategy used during pretraining while ensuring that the total duration covered by the sampled frames remained consistent with the V-JEPA setup. We utilized the Adam optimizer with a weight decay of 0.1 for Ego-Exo4D and 0.01 for FineDiving and JIGSAWS. The learning rate was set to 1e-5 for Ego-Exo4D and JIGSAWS and 5e-5 for FineDiving. We train all models for 300 epochs.

## A.3 ADDITIONAL DATA DETAILS

We provide statistics (number of samples and average video duration) for all dataset tasks in Table 4.

Table 4: Details about dataset tasks. Note that the JIGSAWS suturing and needle-passing tasks are only used for training and not evaluation.

| Dataset | Task | #Samples | Avg. Dur. |
|---|---|---|---|
| Ego-Exo4D (Cooking) | Cooking an Omelet | 44 | 7.24m |
| | Cooking Tomato & Eggs | 28 | 16.3m |
| | Cooking Scrambled Eggs | 20 | 7.21m |
| | Making Cucumber & Tomato Salad | 55 | 3.82m |
| | Making Sesame-Ginger Asian Salad | 32 | 14.71m |
| | Cooking Noodles | 41 | 18.83m |
| | Making Milk Tea | 34 | 5.74m |
| | Making Coffee Latte | 16 | 6.79m |
| FineDiving | Forward 0.5 Som.Pike | 22 | 9.44s |
| | Forward 1.5 Soms.Pike | 36 | 8.79s |
| | Forward 3.5 Soms.Pike | 324 | 9.08s |
| | Forward 3.5 Soms.Tuck | 16 | 10.35s |
| | Forward 4.5 Soms.Tuck | 158 | 9.30s |
| | Back 0.5 Som.Pike | 68 | 8.54s |
| | Back 2.5 Soms.Pike | 204 | 8.45s |
| | Back 2.5 Soms.Tuck | 16 | 7.37s |
| | Back 3.5 Soms.Pike | 72 | 8.58s |
| | Back 3.5 Soms.Tuck | 159 | 8.60s |
| | Reverse 0.5 Som.Pike | 87 | 8.66s |
| | Reverse 2.5 Soms.Pike | 84 | 9.27s |
| | Reverse 1.5 Soms.Tuck | 56 | 8.81s |
| | Reverse 3.5 Soms.Tuck | 233 | 8.74s |
| | Inward 0.5 Som.Pike | 36 | 8.49s |
| | Inward 1.5 Soms.Pike | 24 | 8.10s |
| | Inward 2.5 Soms.Pike | 133 | 8.08s |
| | Inward 2.5 Soms.Tuck | 16 | 7.66s |
| | Inward 3.5 Soms.Tuck | 347 | 8.18s |
| | Arm.Back 3 Soms.Pike | 25 | 8.90s |
| | Arm.Back 3 Soms.Tuck | 53 | 8.51s |
| | Back 0.5 Twist 1.5 Soms.Pike | 12 | 8.49s |
| | Back 1.5 Twists 2.5 Soms.Pike | 209 | 8.59s |
| | Back 2.5 Twists 2.5 Soms.Pike | 79 | 8.57s |
| | Reverse 3.5 Twists 1.5 Soms.Pike | 23 | 8.88s |
| | Reverse 1.5 Twists 2.5 Soms.Pike | 31 | 8.93s |
| | Arm.Fwd 1 Twist 2 Soms.Pike | 10 | 8.49s |
| | Arm.Back 1.5 Twists 2 Soms.Pike | 107 | 8.37s |
| | Arm.Back 2.5 Twists 2 Soms.Pike | 32 | 9.00s |
| | Forward 2.5 Soms.Pike 1 Twist 2.5 Soms.Pike | 94 | 9.19s |
| | Forward 2.5 Soms.Pike 2 Twists 2.5 Soms.Pike | 114 | 8.80s |
| | Forward 2.5 Soms.Pike 3 Twists 2.5 Soms.Pike | 38 | 9.23s |
| JIGSAWS | Knot-Tying | 36 | 57.26s |
| | Suturing | 39 | 1.88m |
| | Needle-Passing | 28 | 1.81m |