# OpenReview forum: "Weakly Supervised Understanding of Skilled Human Activity in Videos"
_ICLR.cc/2025/Conference — ICLR 2025 Conference Withdrawn Submission_

### Official Review · Reviewer_daUo · 2024-10-21

**Soundness:** 3
**Presentation:** 3
**Contribution:** 3
**Rating:** 5
**Confidence:** 3

**Summary:**

The paper proposes a novel method to perform proficiency estimation from long-form videos. In particular, participants are asked to only provide a binary label (proficient or not proficient), which is used to train embeddings of video segments by means of contrastive learning. From the learned embeddings, linear classifiers can then be used to perform the actual proficiency classification and their classification score is used to interpolate between proficient and not proficient. Further, a transformer architecture is utilized to attend specifically to video segments that are indicative of high or low proficiency.

**Strengths:**

1. Proficiency estimation is indeed an important task, especially in educational applications of machine learning. Psychomotor skill learning is an emerging topic, as well.
2. Simplifying annotation of proficiency for human annotators is a worthy goal which may improve inter-rater agreement and reduce the cost of annotations, which is particularly severe for video annotations.
3. The emphasis placed on sparsity of indicators for skill/lack thereof strikes me as important for the field.

**Weaknesses:**

1. I am not quite convinced that binary labeling substantially reduces the annotation problem. Annotators will still have a hard time classifying ambiguous or mixed demonstrations. Rather, it may be more plausible to switch to an annotation framework where erroneous/flawed or skilled parts of a video can be highlighted and the expertise of the entire demonstration can be inferred from there. This type of annotation would also be in line with the sparsity of indicators argued in the paper.
2. The novelty of the approach is not quite clear and should be articulated more explicitly in the introduction.
3. The empiric results only seem partially convincing. In particular, the heading 3.6 "SPARSE SKILL EXTRACTOR APPROACHING FULLY SUPERVISED PERFORMANCE" seems misleading given the rather big gap in performance.
4. The writing left quite a few questions open for me (see below).

**Questions:**

1. Why are both a contrastive learning framework and linear classifiers needed? Wouldn't one suffice?
2. If the linear classification loss does not influence the gradient (due to stop_gradient) - why is it included in the loss in the first place? Couldn't the learned representations be collected at the end of training and a linear classifier be learned from those in a convex optimization setup?
3. Are the contrastive samples selected from the same activity or across activity - i.e. is the network trained to map proficient samples from different activities to the same point? (and similarly for non-proficient samples) Why is it plausible to collapse the variance across activities like this?
4. Similarly: are linear classifiers trained across activities or separately for each activity? If the former: might the latter improve performance?
5. Why were fully supervised models not also evaluated on the the Ego-Exo4D data set?

I would appreciate clarifications which would help me to adjust my score and confidence.

---

### Official Review · Reviewer_iXiK · 2024-10-29

**Soundness:** 2
**Presentation:** 2
**Contribution:** 3
**Rating:** 5
**Confidence:** 3

**Summary:**

The authors propose a model that learns scoring representations using binary proficiency labels through contrastive learning, eliminating the need for human-annotated numerical scores. Additionally, their model can identify specific video segments that indicate low proficiency, enhancing interpretability and providing insights into performance.

**Strengths:**

(1) The idea of addressing a scoring (regression) task using binary labels is effective and provides a novel perspective.

(2) The use of a contrastive learning framework is well-suited to this problem and enhances the model's ability to learn meaningful skill-based representations.

(3) The lightweight token sparsification module effectively identifies and emphasizes segments indicating low proficiency, contributing to model interpretability.

(4) The chosen baseline and dataset are well-suited to evaluating the proposed method, providing a solid foundation to test its effectiveness and showcase its advantages.

**Weaknesses:**

(1) The proposed method is mainly applicable to human activities with procedural elements. In other words, the attention module is limited to time segments. Extending the attention module to the spatial domain could improve its ability to identify indicators of low proficiency in both time and space.

(2) The error recall rate (0.365) in table 2 is not ideal, and a thorough error analysis is needed to better understand and address the model's limitations.

(3) In Table 2, the baseline model V-JEPA achieves a binary F1 score of 0.591, which is close to—and in some cases surpasses—some ablation models. More detailed comparisons are necessary to clarify why the proposed solution outperforms alternatives and to highlight its specific advantages.

**Questions:**

(1) At line 243, it states "given a low proficiency demonstration with error steps." Why does the model focus only on identifying error steps? Would it also be beneficial to locate segments that indicate high proficiency?

(2) In Table 2, the baseline model V-JEPA achieves a binary F1 score of 0.591, which is very close to or even better than some ablation models. What accounts for this similarity in performance, and why does V-JEPA perform competitively with the proposed ablation models?

---

### Official Review · Reviewer_NF5A · 2024-11-02

**Soundness:** 3
**Presentation:** 3
**Contribution:** 3
**Rating:** 6
**Confidence:** 4

**Summary:**

This paper presents Sparse Skill Extractor, a novel framework for understanding skilled human activity in videos using only binary proficiency labels (high or low). The authors argue that this weak supervision approach is advantageous as binary labels are easier to collect, less prone to subjectivity, and can be derived from inherent group characteristics (e.g., expert vs. novice). The core idea of Sparse Skill Extractor is to identify and focus on key moments within a video that are indicative of proficiency, rather than treating all segments equally. This is achieved through a multi-scale contrastive learning approach that incorporates a token sparsification module to prune irrelevant segments, and applies contrastive losses to both local segment features and a global feature generated from the remaining informative segments.

The authors evaluate their approach on three diverse datasets: Ego-Exo4D (cooking tasks), FineDiving (Olympic diving), and JIGSAWS (surgical tasks). Their results demonstrate that Sparse Skill Extractor outperforms weakly supervised baselines on the Ego-Exo4D dataset in both binary and numerical proficiency prediction. The model effectively identifies segments containing annotated errors in low proficiency demonstrations, highlighting its interpretability. Furthermore, Sparse Skill Extractor achieves performance approaching that of fully supervised methods on FineDiving and JIGSAWS, despite using significantly less supervision. Finally, the study explores using experience as a proxy for proficiency on the JIGSAWS dataset, showing promising results for predicting numerical proficiency. The authors conclude that binary proficiency supervision is a viable and efficient approach for developing models with a nuanced understanding of skill.

**Strengths:**

1) The paper introduces a well-crafted framework for assessing proficiency in complex human activities through weak supervision, which is particularly beneficial in domains where precise, frame-by-frame labeling is impractical. By using contrastive learning and token sparsification, the authors achieve competitive results with reduced supervision, making this approach valuable for scalable, real-world applications. The inclusion of interpretability-focused modules, like token sparsification, is particularly impactful. This feature improves model usability, allowing practitioners to understand which parts of a video contribute most to proficiency predictions

2) The authors rigorously test their model on a diverse set of tasks—cooking, diving, and surgical procedures—using three distinct datasets. This comprehensive evaluation demonstrates the model’s versatility across varied task structures and durations, supporting the generalizability of the approach. The use of task-specific datasets also provides insights into the model’s adaptability to both procedural and athletic tasks.

3) By reducing reliance on dense labels, this method opens up pathways for scalable skill assessment, potentially transforming how proficiency is tracked and guided across a variety of domains.

4) The paper includes ablation studies on the local and global contrastive losses and other components, shedding light on the relative importance of each element in the model’s architecture. These experiments provides insights into the model’s internal workings, offering guidance on potential trade-offs for future model adaptations.

**Weaknesses:**

1) The paper uses a fixed sparsification threshold (0.5) without exploring adaptive methods that might allow for task-specific or context-dependent sparsification. An adaptive sparsification mechanism could dynamically adjust based on task complexity or video length, potentially improving interpretability and performance.

2) The paper does not explicitly address granularity within tasks, such as assessing nuanced proficiency in individual sub-steps or segments of a task, beyond the binary classification. While the approach identifies important segments contributing to overall proficiency, it could be strengthened by further differentiating skill levels within sub-tasks. This could allow the model to assess proficiency with greater granularity within a single task.

3) While the paper focuses on segment-based evaluation through token sparsification, it does not consider the sequential and skill-specific transitions that occur within complex tasks, such as moving between sub-steps in a surgical procedure or phases in cooking. Therefore,  detecting these transitions, or examining how skill requirements may shift between task phases, remains a potential area for improvement and could enhance the model’s interpretability and adaptability to nuanced skill variations within tasks.

**Questions:**

1) To what extent do different sparsification levels impact interpretability, and can the authors provide visualizations demonstrating the effects of various sparsification thresholds? By examining different thresholds, the authors could reveal whether a specific sparsification level optimally balances interpretability and prediction accuracy. This analysis could be insightful for understanding the trade-offs and robustness of the sparsification module in the Sparse Skill Extractor.


2) Does the choice of frame sampling interval impact proficiency prediction in both short- and long-form tasks? Could varying this parameter provide insights into an optimal temporal resolution for skill identification?


3) How effectively does the model differentiate between subtle errors in low-proficiency segments across tasks? It would be interesting to see if additional supervision can help in cases where certain skill levels are difficult to distinguish? For instance, an in-depth analysis of attention weights and error detection performance across these segments may highlight specific areas where additional annotations or weak supervision might enhance the model’s sensitivity to nuanced skill distinctions?

4) To what extent does the effectiveness of local versus global contrastive loss vary across different task types (e.g., cooking, diving, surgical tasks)? Would inclusion of further task-specific ablations on these loss components reveal if one is more critical for capturing proficiency-related features within certain task structures?

---

### Official Review · Reviewer_bpbE · 2024-11-04

**Soundness:** 2
**Presentation:** 2
**Contribution:** 2
**Rating:** 3
**Confidence:** 3

**Summary:**

This paper presents Sparse Skill Extractor, a weakly supervised contrastive learning framework for predicting proficiency of human activity skills. Sparse Skill Extractor includes a decision mask for token sparsification, and local and global contrastive loss to contrast high and low proficiency demonstrations. Using three different datasets, it has been compared against supervised and weakly supervised methods for predicting binary and numerical proficiency levels. The proposed method includes a clear method and problem, demonstrating the approach on multiple datasets and baselines. However, the evaluation lacks sufficient rigor to test the method thoroughly. Therefore, I am inclined to reject the paper in its current form but would be open to revisiting this decision if the concerns and questions are addressed comprehensively during the rebuttal process.

**Strengths:**

The method and problem in the paper are written and conveyed in the paper. The method has been demonstrated with 3 datasets and compared with 10 different methods.

**Weaknesses:**

While the Sparse Skill Extractor proposes a new direction, concerns remain regarding the rigor of its presentation, clarification, and evaluation.

1. The general presentation of the paper needs improvement. Clarification of the points that have been questioned in the Weakness section and Question section of this review would be expected.
2. Robustness: The evaluation in Tables 2 and 3 lacks a confidence interval, lacking an evaluation of the robustness of the proposed method.
3. Ablation studies on removing $\mathcal{L}_{global}$ should be performed in Table 2 to see the effect of the term.
4. The result in Table 3 doesn’t say anything about the performance being “promising”. To highlight efficiency in the amount of labels like what has been mentioned in line 434, additional ablation studies on the amount of labels should be performed. Common approaches include:
- Training the model with different percentages of labeled data (e.g., 10%, 20%, 50%, and 100%) and plotting its performance.
- Use varying levels of label granularity for training, such as coarse labels (current, whole-video level) and Finer-grained labels (e.g., segment or frame-level labels).
5. Additional evaluation on varying sparsification levels and pooling layers with varying kernel sizes would be expected to see the effect of sparsification.
6. How has TimeSformer, a supervised method, been adopted to be a weakly supervised method in Table 2? To support the claim of Table 2, other weakly supervised methods, that leverage the pseudorabies, for instance, could be added as a baseline.
7. Qualitative evaluation in Figure 2 is unclear as to how high prediction importance could be connected to “error regions” mentioned in the figure caption. The first darkest blue bar in the first example (medium-level women) corresponds to the error region (doesn’t whisk her eggs fully), but the next darkest blue bar doesn’t correspond to any error region and lacks explanation. This is the same for the male example below, where all three highest importance bars don’t correspond to errors of the performer (“too few tomatoes to cucumber” doesn’t seem to necessarily be related to the error of the performer.

**Questions:**

1. It is unclear how labels are defined with each dataset; for instance, how has the non-procedure domain been defined in Ego-Exo4D? How has the experience been used to label binary labels for proficiency? Was experience level originally “Novice/Intermediate/Expert”?
2. How is the proficiency label used for constructing views of contrastive loss ($y$ and $y’$ in line 136) different than the one used for the final classification loss (lines 211-212)?
- If those are the same labels, then why are fully supervised models not using the binary proficiency predictions to be evaluated using the F1 score in Table 3? Assuming all models use the same test set to evaluate the performance of proficiency prediction, F1 is expected to be calculated on the supervised model as well.
- In the same vein, is there any reason why Error Recall was not calculated with TimeSformer which uses transformer architecture?
3. Is there any reason why all three datasets have been used for evaluation in Tables 2 and 3? (e.g., why has Ego-Exo4D only been used for evaluation against weakly supervised methods?)
4. What was the predefined target ratio $\mu$ for each dataset?

---

### Note · Authors · 2024-11-25

**Comment:**

We greatly appreciate the reviewers’ thoughtful and constructive feedback. After careful consideration, we have decided to withdraw the paper so we can fully address all comments and further improve our work.

**Withdrawal Confirmation:**

I have read and agree with the venue's withdrawal policy on behalf of myself and my co-authors.